# The Geochemistry of 1 ky Old Euxinic Sediments of the Western Black Sea

**Octavian G. Duliu** [1,2,*,†] **, Carmen I. Cristache** [3,†] **, Ana-Voica Bojar** [4,5,†] **, Gheorghe Oaie** [6,†] **, Otilia-Ana Culicov** [2,7,†] **, Marina V. Frontasyeva** [2,†] **and Emil Constantinescu** [8,†]

1   Department of the Structer of Matter, Earth and Atmospheric Physics and Astrophysics, University of Bucharest, P.O. Box MG-11, 077125 Magurele (Ilfov), Romania
2   Joint Institute for Nuclear Research, Joliot Curie str. 6, 141980 Dubna, Russia; culicov@nf.jinr.ru (O.-A.C.); marina@nf.jinr.ru (M.V.F.)
3   National Institute of Research and Development for Physics and Nuclear Engineering "Horia - Hulubei", P.O. Box MG-6, 077125 Magurele (Ilfov), Romania; carmen.cristache9@gmail.com
4   Geographie und Geologie, Salzburg University, Kapitelgasse 4–6, 5020 Salzburg, Austria; Ana-Voica.Bojar@sbg.ac.at
5   Studienzentrum Naturkunde, Universalmuseum Joanneum, Weinzöttlstraße 16, 8045 Graz, Austria
6   National Institute of Marine Geology and Geoecology, Dimitrie Onciul str. 23–25, 024053 Bucharest, Romania; astanica@geoecomar.ro
7   National Institute for Research and Development in Electrical Engineering, 313 Splaiul Unirii, 060032 Bucharest, Romania
8   Department of Mineralogy, University of Bucharest, Nicolae Balcescu Blv. 1, 010041 Bucharest, Romania; emil@constantinescu.ro
*   Correspondence: o.duliu@upcmail.ro
†   These authors contributed equally to this work.

**Abstract:** To get more data on the geochemistry of Black Sea euxinic sediments, a 50-cm core was collected at a depth of 600 m on a Western Black Sea Continental Platform slope. The core contained unconsolidated sediments rich in cocoolithic ooze and mud. Epithermal Neutron and Prompt Gamma Activation Analysis were used to determine the content of nine major (Na, Mg, Al, Si, K, Ca, Ti, Mn, and Fe as oxides) and 32 trace elements (Cl, Sc, V, Cr, Co, Ni, Zn, As, Se, Br, Rb, Sr, Zr, Mo, Sn, Sb, Cs, Ba, La, Ce, Nd, Sm, Eu, Gd, Tb, Dy, Yb, Hf, Ta, W, Th, and U) with a precision varying between 3 and 9%. The core contained unconsolidated sediment rich in coccolithic ooze and mud. Previous $^{210}$Pb geochronology suggests an age of ∼1 ky of considered sediments. Major components distribution showed that, except for Cl and Ca, the contents of all other elements are similar to Upper Continental Crust (UCC) and North American Shale Composite (NASC). The distribution of the 32 trace elements showed similarities to the UCC, except for redox-sensitive metals Fe, Se, Mo, and U, of which the significantly higher content reflects the presence of euxinic conditions during deposition. A chondrite normalized plot of nine rare earth elements indicated a similarity to UCC and NASC, suggesting a continental origin of sedimentary material.

**Keywords:** Black Sea; geochemistry; unconsolidated sediments; euxinic zone; redox-sensitive elements

## 1. Introduction

The Black Sea, an inland sea with a surface area of 422,000 km$^2$ and a maximum depth of 2212 m [1], is also the largest meromictic basin in the world. About 7 ky ago, after a massive influx of the Mediterranean seawater through the Bosporus strait, the Black Sea stratified into two unmixed layers: a surface layer, well-oxygenated one with a thickness varying between 120 and 180 m, and

a much thicker layer below the oxygenated one, filling the rest of the basin. The lower layer, devoid of oxygen and saturated with hydrogen sulfide, forms an euxinic environment, populated only by extremophile bacteria that produce hydrogen sulfide and carbon dioxide [2,3].

As the euxinc zone is completely devoid of bioturbating organisms, it represents an ideal environment to preserve sedimentary structure for long periods of time, thus allowing a systematic reconstruction of past processes.

The Black Sea catchment basin extends over Europe and the Anatolian Peninsula and covers an area of 1,874,904 $km^2$. The main tributaries—Danube, Dniester, Bug, and Dnepr—with a combined inflow of 261 $km^3$/y, represent 76 % of the entire tributary discharge into the Black Sea [4]. The Danube itself contributes annually with about 3.0–3.5 $\times$ $10^7$ tons of sediments, which are discharged and spread over the western continental platform [5,6] as well as within the euxinic zone.

We present and discuss our experimental results concerning the elemental composition of the unconsolidated marine sediments collected from the euxinic zone of the Black Sea. Detailed vertical profiles of of redox-sensitive metals, such as Fe, Se, Mo, or U are useful for the reconstruction of past environmental events [5–13], while the distribution of other elements such as Sc, Zr, REE(rare earth elements), Ni, and Th can provide relevant information concerning the source of sedimentary material [14–17].

In order to assess the past geochemical evolution as reflected in a 50-cm-long core, of which estimated age was about 1 ky [18], and to increase the accuracy of measurements, we have used two complementary analytical techniques: Epithermal Neutron Activation Analysis (ENAA) and Prompt Gamma-ray Activation Analysis (PGAA).

As the main objects of this study aim to obtain additional data concerning the possible evolution of the euxinic environment in the past 1 ky as well as the origin of sedimentary material, we have focussed our analysis on the most important elements for this study, i.e., Fe and redox-sensitive elements Se, Mo, and U as well as Sc, Ni, Zr, eight REE, and Th.

## 2. Materials and Methods

### 2.1. Samples

A 50-cm-long core, (code number BS 600) was collected at a water depth of 600 m using a Mark II-400-type multicorer during a June 2004 scientific cruise of the National Institute of Marine Geology and Geoecology R/V Mare Nigrum [19]. The collecting point was located eastward of Constanta (Romania) (Figure 1, inset). After collection, the 12-cm diameter BS 600 core was sealed and stored in vertical position in a refrigerator for about a week. Then, after the end of cruise, the core was examined by means of a Siemens Somatom HQ Computer Tomograph, which showed an alternation of about 254 horizontal-millimetres thick and almost parallel laminae of coccolithic and argilaceous mud, undisturbed by biotic activity (Figure 1a,b).

A few weeks later, the core was longitudinally split into two halves, with one of them being divided into 45 segments, 5 mm to 5 cm thick, for further investigations [20]. Each segment was then dehydrated at 105 °C, homogenized, and divided into four aliquots, with two of them of about 1 g for ENAA and PGAA, one of them of 15 to 100 g for radiometric, and one of them of about 10 g for additional mineralogical analysis and Total Organic Carbon (TOC) determination.

### 2.2. Analytical Techniques

Instrumental Neutron Activation Analysis (INAA) still represents one of the best analytical methods to determine the concentration of 38 to 40 elements existing in nature in a range of concentrations up to 100 ng/kg without laborious radiochemical separations or acid digestion [21,22].

ENAA was performed at the IBR-2 pulsed fast reactor of the Joint Institute for Nuclear Research (JINR), Dubna (Russian Federation), and PGAA was performed at the Budapest Research Reactor, Hungary. Certified reference materials IAEA-SL-3 lake sediment and IAEA SL-7 soil were used for

calibration, while IAEA SL-1 lake sediment was used to check the accuracy of measurements. All certified reference materials were irradiated together with the investigated samples under the same conditions. Except Na, the content of all elements as determined by both methods were coincident at $p < 0.05$. For all elements, the precision in determining their concentrations was better than 9% (Table A1).

In the case of ENAA, the neutron energies varied between thermal 0.025 eV and resonant 500 eV which significantly increases both sensitivity and accuracy of measurement [23]. The PGAA, which is based on the analysis of the gamma rays emitted during neuron capture by nuclei, allows determining with high accuracy the content of some elements such as Gd for which classic INAA gives great errors. More details concerning ENAA and PGAA determinations can be found in References [24–26].

The mineralogical analysis of major components was performed by X-ray diffraction (XRD) using a Cu anode DRON-3 diffractometer provided with a graphite monochromator and a NaI(Tl) detector, while the presence of trace minerals was investigated by microscopy using a binocular microscope. The XRD spectra were interpreted by means of the Crystallography Open Database [27] .

Finally, TOC was determined using a LECO RC-412 Carbon Analyzer (Leco.inc., Saint Joseph, MI, USA) with an accuracy of about 10%.

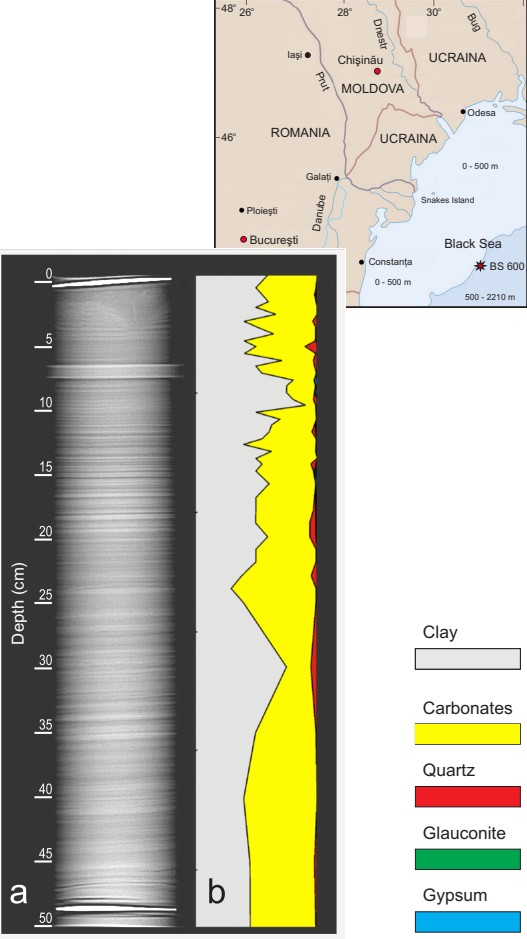

**Figure 1.** A tomographic image of the BS 600 core (**a**) and the vertical distribution of the main mineral fractions (in %) (**b**): The inset illustrates the position of the BS 600 collecting point location on the slope of the Western Black Sea continental platform.

## 3. Results

The vertical distribution of the main minerals, as resulted by X-ray diffraction determination, is illustrated in Figure 1b, while the corresponding numerical values of their concentrations are provided

in Table A2. A careful examination of both Figure 1b and Table A2 showed that the clay and carbonate, the main mineral fractions, account for about 98 % while quartz, gypsum, and glauconite are the minor ones. It should also be noted that, in the upper portion of the core, carbonate decreases while quartz was more abundant (Figure 1b). Traces of other minerals such as titanite, rutile, and zircon as well as microorganism tests (mainly *Emiliania huxlei* cocoolithes) were identified under binocular microscopy. The TOC content varied between 3% and 6%, in good concordance with previous data reported in Reference [28].

The final experimental data regarding the content of the nine major oxides (in wt %) and 31 trace elements (in mg/kg) as determined by ENAA and PGAA can be freely accessed at http://dx.doi.org/ 10.17632/d5f8c9gtv7.2 [20], while the main statistical descriptors inclusive the experimental uncertainty defined as the combination of statistical and certified material uncertainties are provided in Table A1.

## 4. Discussion

Previous analysis of this core demonstrated that, by assuming a constant depositional rate, the maximum age of the sediments is about $1 \pm 0.1$ ky [18]. Given the quasi-regularity of lamina thickness (Figure 1), we have used the $^{210}$Pb data for determination of an estimated age to different sections of the core. At the same time, the core digital radiography has revealed the presence of about 254 laminae of which width varied between 1 and 3 mm (Figure 1), so we have attributed the core sediments to the Stratigraphic Unit 1 [29]—the coccoliths mud [28,29].

The carbonate-rich coccolithic mud is well evidenced by the spider diagram (Figure 2a) which illustrates the distribution of eight major elements normalized to UCC [30] as well as on the $SiO_2 - Al_2O_3 - Na_2O + K_2O + CaO$ ternary diagram shown in Figure 2b.

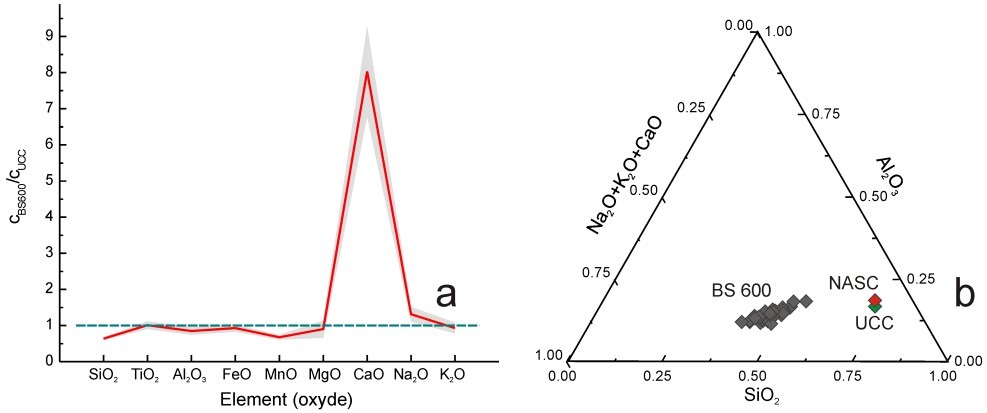

**Figure 2.** Spider diagram illustrating the distribution of the major elements (as oxides) normalized to Upper Continental Crust (UCC) (**a**) as well as a $SiO_2 - Al_2O_3 - Na_2O + K_2O + CaO$ ternary diagram (**b**) illustrating the relationship between the BS 600 major elements and the corresponding UCC and North American Shale Composite (NASC) ones.

According to the spider diagram, with the exception of CaO, the content of all other major oxides remained relatively closer to the UCC (Figure 2a). This observation suggests that the BS600 sedimentary material is closer to the UCC with the exception of CaO, which appears $7.9 \pm 1.2$ times higher then the UCC one (Table A2). In our opinion, the increased content of CaO reflects a carbonate accumulation, most probaby due to the presence of *Emiliania huxleyi* coccoliths [29,31] (Table A1).

This peculiarity is also evidenced on the ternary diagram illustrated in Figure 2b. Indeed, on this diagram, all BS600 sediments are shifted towards $SiO_2$ smaller values with respect to UCC [30] and NASC [32] points. All of them form an elongated ellipse, so that the higher the CaO content, the more distant with respect to UCC and NASC they appear on the diagram.

In this regard, it should be noted that the iron content as FeO, if normalized to the sum of all other oxides except CaO, gives a value of $0.067 \pm 0.007$, higher than the corresponding UCC value of 0.052. This suggests a moderately enriched content of iron in BS600 sediments in good agreement with literature data for the euxinic basin margin [10] but lower than those recorded for the deep euxinic basin [33].

Concerning the interrelation between the oxides of major components and Cl, the correlation analysis shows the existence of at least two clusters (Table 1). One of them, which consists of Na, K, and Cl, is most probably associated with of interstitial sea water. The content decreases towards core bottom by sediments compaction, a hypothesis sustained by the almost coincident vertical profile of $Na_2O$ and Cl (Figure A1). The second cluster contains Al, Si, Ti, and Fe, presumably connected with the clay. At the same time, the calcium oxide negatively correlates with almost all other oxides as well as Cl, peculiarly consistent with the $CaCO_3$ distribution and totally complementary to the clay, as mentioned before.

**Table 1.** The matrix of Spearman's correlation coefficients between the oxides of major elements and Cl: The correlation significant at $p < 0.05$ are represented in red and green.

|  | $SiO_2$ | $TiO_2$ | $Al_2O_3$ | FeO | MnO | MgO | CaO | $Na_2O$ | $K_2O$ |
|---|---|---|---|---|---|---|---|---|---|
| $TiO_2$ | 0.816 |  |  |  |  |  |  |  |  |
| $Al_2O_3$ | 0.429 | 0.346 |  |  |  |  |  |  |  |
| FeO | 0.572 | 0.503 | 0.799 |  |  |  |  |  |  |
| MnO | −0.133 | −0.093 | 0.555 | 0.378 |  |  |  |  |  |
| MgO | −0.142 | −0.093 | −0.368 | −0.264 | −0.231 |  |  |  |  |
| CaO | −0.839 | −0.638 | −0.636 | −0.783 | −0.027 | −0.045 |  |  |  |
| $Na_2O$ | 0.212 | 0.167 | 0.125 | 0.206 | −0.283 | 0.022 | −0.429 |  |  |
| $K_2O$ | 0.520 | 0.363 | 0.311 | 0.511 | −0.078 | −0.046 | −0.704 | 0.564 |  |
| Cl | 0.211 | 0.149 | −0.026 | 0.186 | −0.149 | 0.036 | −0.438 | 0.889 | 0.432 |

Therefore, the BS 600 sediments appear, from the perspective of rock-forming elements, close to UCC and NASC. This peculiarity points out a predominance of terrigenous materials with a significant proportion of *Emiliania huxleyi* calcium carbonate [29]. At the same time, the moderately elevated Fe content could be attributed to the euxinic environment under which the sediments were deposited [10,33].

According to References [34,35], the particulate Fe oxides that exist in the upper, shallow oxic layer of the Black Sea when entering into anoxyc/euxinic water column are dissolved as Fe(III) is reduced to soluble Fe(II) and when reacting with dissolved hydrogen sulphide is deposited into sediments [10], where its content exceeds the content of the original sedimentary material.

For a better understanding of the behaviour of redox-sensitive elements under the Black Sea euxinic environment, in Figure 2, we have reproduced the vertical profile of the corresponding authigenic enrichment factors (*EF*) [33] defined as

$$EF = \left[\frac{c_x}{c_{Al}}\right]_{BS600} \bigg/ \left[\frac{c_x}{c_{Al}}\right]_{UCC} \tag{1}$$

where the first term of the equation represents the ratio of the contents of considered element $x$ to Al, as a descriptor of the detrital component of the sediments [28], while the second term is assigned to the ratio of the same elements in a reference material, in our case, the UCC. $x$ stands for Se, Mo, U, and Fe.

We have considered the UCC as a reference material by taking into account the previous results concerning the content of the nine major elements which indicate, for the BS600 sediments, a significant resemblance to both UCC and NASC average detrital materials.

Based on the experimental data [20], we have determined *EF* values that varied between 2.4 and 170 in the case of Se, Mo, and U and between 0.75 and 1.36 with an average value $1.19 \pm 0.13$ in the case of Fe (Table 2). By comparing our data with the literature ones [10,12,33], we noticed that, in the present case, the enrichment factors, except for in Reference [12], were slightly smaller.

**Table 2.** The numerical values of the enrichment factor (*EF*) of the investigated redox-sensitive elements in BS600 sediments and literature data.

| Element | *FF* | Literature Data | | Element | *FF* | Literature Data | |
|---------|------|-----------------|---|---------|------|-----------------|---|
| Fe | 0.75–1.36 | 0.93–1.28 euxinic margin | [10] | Mo | 28–60 | 80–376 euxinic stations 9 and14 | [33] |
| | | 1.84–2.78 euxinic basin | | | | | |
| | | 1.2–1.7 euxinic stations 9 and 14 | [33] | | | | |
| Se | 12–170 | <59 clayed ooze | [12] | U | 2.4–6.1 | 1.5 clayed ooze | [12] |

We explained this finding by the position of the BS600 collecting point on the slope of the continental platform, while the literature data [10,12,33] refer to the Black Sea abyssal plain where the euxinic water column is significantly thicker.

This assumption is sustained in the case of Fe of which the content (Table A1) as well as *EF* are comparable with similar values provided in Reference [10] for the euxinic margin but smaller than those reported for the euxinic basin [33]. In the last case, all samples were collected in two deep stations located in the central part of the Black Sea where the euxinic water column reaches maximum values. In a similar situation, we have found in the case of Mo, of which *EF* reached a maximum value of 60, a smaller value than the minimum value of 80 reported in Reference [33].

This situation could also contribute the finely suspended sedimentary material transported by the Danube river that could reach the BS600 sampling point situated on the slope of the continental platform at about 200 km from the Daube Delta (Figure 1—inset).

Another peculiarity observed for the *EF* vertical profiles (Figure 3) is the significant fluctuation at depths varying between 15 to 25–30 cm, the same depths where the clay and carbonate contents reached the maximum variability (Figure 1b). By assuming for the entire column an age of about 1 ky [18], this sector could have an age between 300 and 600 y BP, which partially coincides with the Little Ice Age. Another possible explanation could be linked with the onset of the Industrial Revolution and its significant environmental impact.

A similar peculiarity was evidenced by investigating the vertical distribution of the La/Th ratio. Usually, its value varies between 2.85 for loess [36] and 2.97 for UCC [30]. In our case, we have noticed an average value of 2.95 ± 0.29, consistent with UCC. After a careful examination of its vertical distribution along the sediment column, we noticed that this ratio is almost constant between 20 and 50 cm and equal 2.8 ± 0.14 but increases to the surface of sediments to almost 3.5 (Figure A3a). We have noticed an analogous behaviour for the La/Yb ratio. Although the average value of the La/Yb ratio was of 13.4 ± 2.1, very close to 13.6, which is characteristic for the UCC [36], the La/Yb vertical profile showed increasing values towards the sediment surface (Figure A3b) mainly due to a sizeable enlargement of the La content while Yb and Th content remained almost constant along the entire core [20].

Both cases showed a remarkable resemblance to the vertical profiles of anthropogenic Zn, As, Br, Sn, and Sb reported earlier for the same sediments [37] and attributed to the 18th to 20th century industrial development of the Western European countries accompanied by massive deforestation. At the same time, factories and power plants frequently located near water bodies disposed large amounts of industrial waste, which were transported by tributary rivers and especially by Danube, and have contaminated the superficial layers of the Black Sea sediments.

While the *EF* vertical profiles of Fe, Mo, and U fluctuated at about the same value, the Se *EF* presents a well-defined tendency to increase with the depth (Figure 3b).

Selenium belongs to the 16th group of the periodic table. In sea water and under oxic conditions, Se is soluble as selenate $SeO_4^{2-}$ or selenite $SeO_3^{2-}$ ions [38]. In a more reducing environment such

as an euxinic one, Se occurs as insoluble selenide $Se^{2-}$ ion or even as metallic Se, which are further incorporated into sediments [39]. Moreover, the presence of insoluble Se is related to the activity of both aerobic and anaerobic organisms. Therefore, the presence of the euxinic environment together with the activity of anaerobic bacteria could explain the increased content of Se in BS600 sediments as well as its characteristic vertical profile (Figure 3b).

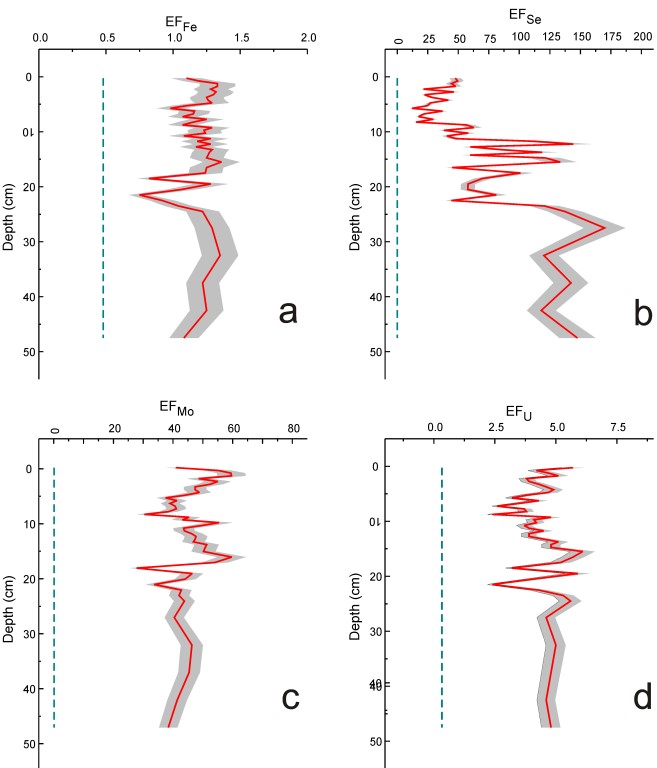

**Figure 3.** The vertical distribution of the *EF* of Fe (**a**), Se (**b**), Mo (**c**), and U (**d**): The vertical dashed lines correspond to the UCC enrichment factor.

A further Principal Component Analysis (PCA) performed on all trace elements within all 45 sections suggested the presence of two clusters. The first cluster consists of the sediments up to a depth of 20 cm, and the other one comprises the rest of sediments until the core bottom (Figure A1).

The loading factors of the Principal Components (PCs) PC1 and PC2 are reproduced in Table A3 (Appendix A). In the case of PC1 which assures 38.5% of total variability, the main contributors are $Na_2O$, Cl, and $K_2O$, i.e., elements associated with the marine environment and which, as mentioned before, form a cluster (Table 1). The PC1 contains also the possible anthropogenic contaminants Zn, As, Br, Sn, and Sb as well La, i.e., those element of which content significantly increased toward sediment surface [37] (Figure A3). It should be remarked that the reduced contribution to PC1 of $SiO_2$, $TiO-2$, $Al-2O-3$, FeO, and MgO, i.e., the major of components of sedimentary material of which presence could be attributed to argillaceous fraction, suggests the relative uniformity of the mineralogical composition of BS600 sedimentary material.

In the case of PC2, the main contribution comes from CaO and Sr, two components mainly associated with the carbonate fraction of sediments.

The contents of relative immobile elements such as Ti, a ubiquitous component of the terrestrial rocks and Ni, could be used to differentiate immature sedimentary material resulting from magmatic precursors from the mature sedimentary ones [16]. Correlated with a ternary discriminating diagram Sc-La-Th [36], this type of graphic analysis was useful in confirming the terrigenous origin of BS600 sedimentary material. Indeed, both Figure 4a,b point towards a significant resemblance of the BS600 material with the UCC and NASC.

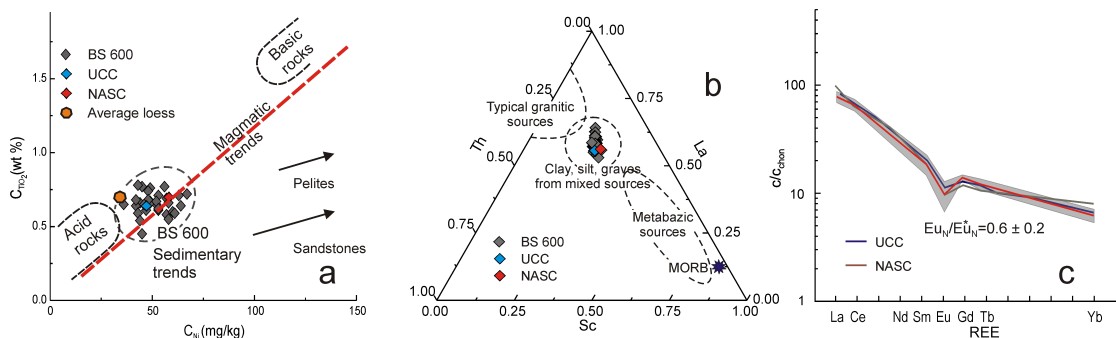

**Figure 4.** A TiO$_2$ vs Ni diagram [16] (**a**), a discriminating ternary Sc-La-La (**b**) [15], as well as a chondrite normalization REE plot (**c**) proving the similarity between the BS600 sediments and UCC [30] and NASC [32], suggesting a continental, terrigenous origin of the BS600 sediments.

This finding could be confirmed by the distribution of eight REE as determined by ENAA (La, Ce, Nd, Sm, Eu, Tb, and Yb) and PGAA (Gd) and normalized to chondrite [40] (Figure 4c). This diagram evidences no negative Ce anomaly as would be expected in the case of euxinic/reducing conditions but a well-represented negative Eu anomaly [15] described by an average value of Eu$_N$/Eu$_{N*}$ of 0.6 ± 0.2, close to 0.67 for UCC [30].

The absence of a Ce negative anomaly could be related to the relative reduced depth of the anoxic/euxinic water column of only 450 m as well as to the vicinity of Danube Delta through which the majority of fresh sedimentary material is discharged into Black Sea. In the absence of more literature data on Ce anomalies, further research concerning different region of the Black Sea would be necessary.

While Ce anomaly is determined by the oxic-anoxic condition of a depositional environment, an Eu negative anomaly is a characteristic for UCC [30] as well as recycled sedimentary rocks [15] such as NASC [32] or PAAS [36]. This fact confirms once more the contribution of terrigenous continental material to BS600 sediments.

Zircon, the natural silicate of zirconium shows a remarkable resilience to abrasion. For this reason, Zr content by means of a Th/Sc vs. Zr/Sc biplot could be used in establishing to which extent the sedimentary material was recycled [41]. In the case of BS600 material, all samples showed a relatively reduced content of Zr (Figure 5), all of them being grouped around the UCC and NASC locations, different from the Dobrogea (Romania) loess [42], in good agreement with Reference [4], which indicates also a negligible fraction of loess in the BS600 core sedimentary material.

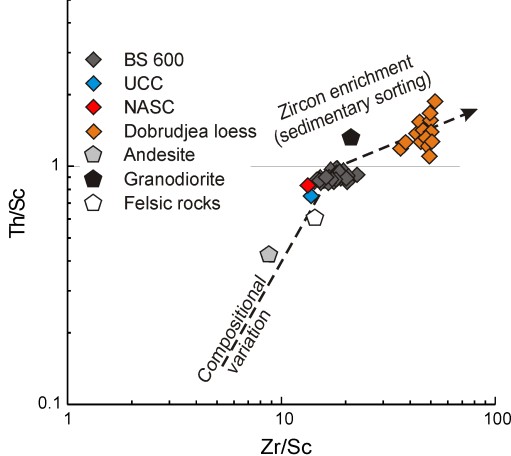

**Figure 5.** A discriminating biplot of Th/Sc vs. Zr/Sc illustrates the absence of Zr enrichment of BS600 sediments due to weak recycling, suggesting the presence of a relative fresh sedimentary material [41].

## 5. Conclusions

The vertical profiles of nine major elements as well as 33 trace elements were investigated in a 50-cm core collected from the euxinic zone of the Western Black Sea by means of instrumental neutron activation analysis. Previous $^{137}$Cs and $^{210}$Pb geochronologies documented an age of about 1000 years for the oldest sediments. As the core was sliced in 45 segments of which thickness varied between 0.5 and 5 cm, it was possible to reconstruct the depositional environment with a resolution of 10 and 100 years.

Our study evidenced a complex sedimentary pattern consisting of both terrigenous material close to the upper continental crust and coccolithic calcium carbonate, all coexisting in a relatively stationary euxinic condition as the vertical profile of redox-sensitive elements Fe, Se, Mo, and U confirm. Their vertical profile suggests changes of the sediments geochemistry for the past four centuries, most probably associated with the end of the Little Ice Age and the Industrial Revolution in Western Europe.

A chondrite-normalized plot of eight REE displayed a Eu negative anomaly, in good concordance with the continental origin of sedimentary material.

**Author Contributions:** Conceptualization, G.O. and O.G.D.; methodology, G.O., O.G.D., C.I.C., O.-A.C., and M.V.F.; statistic analysis, O.G.D.; validation, A.-V.B. and E.C.; writing and editing, O.G.D.

**Funding:** This research received no external funding.

**Acknowledgments:** The work was partially realized within cooperation protocols no. 4322-4-14/16 and 4322-4-17/19 between the University of Bucharest and the Joint Institute for Nuclear Research, Dubna, Russian Federation represented by the Frank Neutron Physics Laboratory. We are much indebted to Dan Secrieru from the National Institute of Marine Geology and Geoecology for his useful comments concerning sediment geochemistry and to Mariana Marinescu for her help in editing the manuscript. Many thanks to Jasmina Obhodas for her interest and support in presenting these data.The work is dedicated to the memory of Gigi Oaie, the former director of the National Institute for Marine Geology and Geoecology, Bucharest, Romania who passed away during the preparation of this manuscript. We are also grateful to the three anonymous reviewers for their useful remarks and suggestions

**Conflicts of Interest:** The authors declare no conflict of interest.

## Abbreviations

The following abbreviations are used in this manuscript:

| | |
|---|---|
| EF | Enrichment Factor |
| ENAA | Epithermal Neutron Activation Analysis |
| INAA | Instrumental Neutron Activation Analysis |
| NASC | North American Shale Composite |
| PAAS | Post-archaean Australian Average Shale |
| PC | Principal Component |
| PCA | Principal Component Analysis |
| PGAA | Prompt Gamma-ray Activation Analysis |
| UCC | Upper Continental Crust |
| XRD | X-ray Diffraction |

## Appendix A

**Table A1.** The average values (*c*), standard deviations (*σ*), and combined standard uncertainty ($u_c$) of BS600 elements together with the corresponding contents of UCC and NASC: Oxide contents are expressed in wt %, and the content of the other elements are expressed in mg/kg. *n*—no data. Total experimental uncertainty is defined as the combination of statistical and certified material uncertainties.

| Element | *c* | *σ* | $u_c$ | UCC | NASC | Element | *c* | *σ* | $u_c$ | UCC | NASC |
|---|---|---|---|---|---|---|---|---|---|---|---|
| $SiO_2$ | 42.07 | 2.56 | 3.0% | 66.62 | 64.8 | Zr | 189 | 27 | 3.8% | 193 | 200 |
| $TiO_2$ | 0.64 | 0.07 | 6.2% | 0.64 | 0.7 | Mo | 42.5 | 5 | 3.3% | 1.1 | *n* |
| $Al_2O_3$ | 12.87 | 1.56 | 2.5% | 15.4 | 16.9 | Sn | 2.9 | 1.4 | 6.7% | 2.1 | *n* |
| FeO | 5.11 | 0.5 | 3.8% | 5.04 | 5.66 | Sb | 2.7 | 1.4 | 8.7% | 0.4 | *n* |
| MnO | 0.07 | 0.01 | 4.9% | 0.1 | 0.06 | I | 94 | 24 | 8.5% | 1.4 | *n* |
| MgO | 2.2 | 0.59 | 5.2% | 2.48 | 2.86 | Cs | 5.5 | 0.7 | 6.3% | 4.9 | *n* |
| CaO | 28.41 | 4.41 | 7.0% | 3.59 | 3.63 | Ba | 488 | 127 | 8.8% | 624 | 636 |
| $Na_2O$ | 4.24 | 0.66 | 7.5% | 3.27 | 1.14 | La | 29 | 4 | 8.5% | 31 | 31 |
| $K_2O$ | 2.56 | 0.42 | 4.5% | 2.8 | 3.97 | Ce | 63 | 8 | 8.3% | 63 | 67 |
| Cl | 4030 | 620 | 3.5% | 370 | *n* | Nd | 22.7 | 6.6 | 7.5% | 27 | 27 |
| Sc | 11 | 1.2 | 6.9% | 14 | 15 | Sm | 3.8 | 0.95 | 5.8% | 4.7 | 5.6 |
| V | 110 | 19 | 8.8% | 97 | 130 | Eu | 0.8 | 0.3 | 8.8% | 1 | 1.2 |
| Cr | 62 | 11.2 | 7.7% | 92 | 125 | Gd | 4.1 | 0.8 | 8.9% | 4 | 5.2 |
| Co | 16 | 2 | 6.8% | 17.3 | 26 | Tb | 0.7 | 0.1 | 8.3% | 0.7 | 0.05 |
| Ni | 53 | 8 | 8.9% | 47 | 58 | Yb | 1.8 | 0.2 | 8.2% | 2 | 3.1 |
| Zn | 82 | 19 | 5.5% | 67 | *n* | Hf | 5.1 | 0.6 | 8.5% | 5.3 | 6.3 |
| As | 11.1 | 2 | 7.9% | 4.8 | *n* | Ta | 0.7 | 0.1 | 7.6% | 0.9 | 1.1 |
| Se | 5.2 | 3.3 | 6.9% | 0.09 | *n* | W | 5 | 3.1 | 7.3% | 1.9 | *n* |
| Br | 87 | 20.8 | 8.9% | 1.6 | *n* | Th | 9.8 | 1.2 | 7.1% | 10.5 | 12.5 |
| Rb | 84 | 11.1 | 8.2% | 84 | 125 | U | 9.9 | 1.2 | 8.3% | 2.7 | 2.7 |
| Sr | 710 | 180 | 5.5% | 320 | 142 | | | | | | |

**Table A2.** The vertical distribution of the content (in %) of main mineral components of the BS600 sediments: Carb—carbonate, Qtz—quartz, Gl—glauconite, and Gyp—gypsum. (G. Caraivan, unpublished results).

| Depth (mm) | Clay | Carb | Qtz | Gl | Gyp | Depth (mm) | Clay | Carb | Qtz | Gl | Gyp |
|---|---|---|---|---|---|---|---|---|---|---|---|
| 0–5 | 34 | 65 | - | - | - | 115–120 * | 35 | 60 | 3 | - | 1 |
| 5–10 | 49 | 50 | 1 | - | - | 120–125 | 44 | 55 | 1 | - | - |
| 10–15 | 43 | 55 | 1 | - | 1 | 125–130 | 60 | 40 | <1 | - | - |
| 15–20 | 39 | 90 | 1 | - | - | 130–135 * | 35 | 60 | <5 | - | - |
| 20–25 | 50 | 50 | - | - | - | 135–140 | 50 | ~50 | <1 | - | - |
| 25–30 | 35 | 65 | - | - | - | 140–145 | 40 | 55 | 4 | - | 1 |
| 30–35 | 56 | 40 | <4 | - | - | 145–150 * | 47 | 50 | 1 | - | 1 |
| 35–40 | 48 | 50 | 2 | - | - | 150–160 | 38 | 60 | ~2 | - | - |
| 40–45 | 32 | 60 | 8 | - | - | 160–170 | 48 | 50 | 1 | - | ~1 |
| 45–50 | 58 | 40 | 1 | - | <1 | 170–180 | 47 | 50 | 2 | - | <1 |
| 50–55 | 58 | 40 | 1 | - | <1 | 180–190 | 45 | ~50 | <5 | <1 | - |
| 55–60 | 58 | 40 | 1 | 1 | - | 190–200 | 35 | ~60 | <5 | <1 | - |
| 60–65 | 25 | 70 | 4 | 0.5 | 0.5 | 200–210 | 49 | ~50 | <1 | - | - |
| 65–70 | 48 | 50 | 2 | - | - | 210–220 | 49 | ~50 | <1 | - | - |
| 70–75 | 45 | ~55 | <0.5 | - | - | 220–230 | 60 | ~40 | <0.5 | - | - |
| 75–80 | 17 | 80 | 2 | - | <0.5 | 230–240 | 70 | ~30 | <1 | - | - |
| 80–85 | 23 | 75 | <1 | - | 1 | 240–250 | 60 | ~40 | <1 | - | - |
| 85–90 | 23 | 75 | 0.5 | - | 1.5 | 250–300 | 20 | 75 | 5 | - | - |
| 90–95 | 16 | 80 | 3 | | <1 | 300–350 | 50 | ~50 | <1 | - | - |
| 95–100 | 8 | 90 | 1 | 1 | - | 350–400 | ~60 | ~40 | - | - | - |
| 100–105 | 45 | 90 | 1 | - | - | 400–450 | 53 | 45 | 2 | - | - |
| 105–110 | 30 | 70 | <0.5 | - | - | 450–500 | 54 | 45 | 1 | - | - |
| 110–115 | 35 | 63 | ~1 | - | 1 | - | - | - | - | - | - |

* 0.5–~1% microorganisms tests.

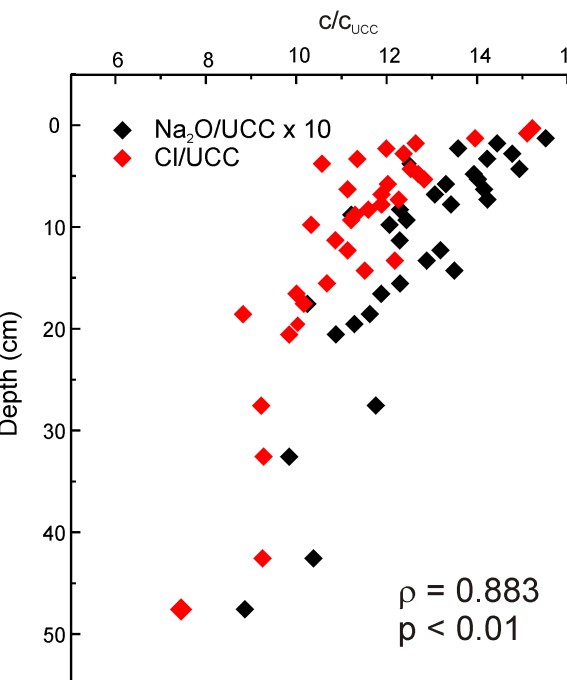

**Figure A1.** The vertical profile of $Na_2O$ and Cl in the BS600 sediments: Both contents were normalized to the UCC [30]. For a better illustration, the $Na_2O$ content normalized to UCC [30] was represented multiplied by 10. It should be remarked that both contents decrease synchronically with the sediment depth.

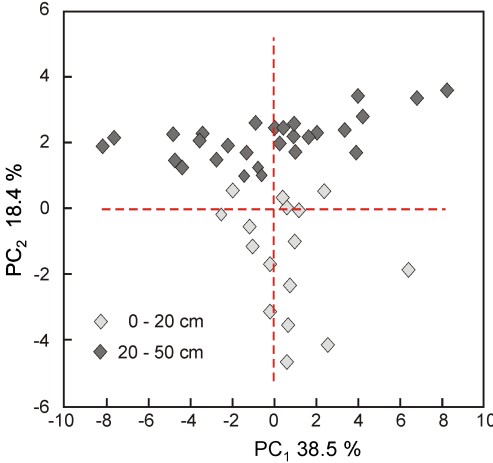

**Figure A2.** The result of the Principal Component Analysis (Q mode) suggesting the existence of two clusters: one consisting of sediments covers the first 20 cm of the DS600 sedimentary column, and the other consists of the rest of the samples. It should be remarked that according to Reference [18] geochronology, the first 20 cm correspond to the last 400 y, the end of the Little Ice Age, and the beginning of the Industrial Revolution in Western and Central Europe.

**Table A3.** The loading factors of the Principal Components (PCs) PC1 and PC2.

| Element | PC1 | PC2 | Element | PC1 | PC2 | Element | PC1 | B |
|---|---|---|---|---|---|---|---|---|
| $SiO_2$ | 0.231 | −0.442 | Ni | 0.104 | 0.005 | La | 0.538 | −0.452 |
| $TiO_2$ | 0.086 | −0.225 | Zn | 0.828 | 0.117 | Ce | −0.123 | −0.239 |
| $Al_2O_3$ | −0.118 | −0.589 | As | 0.706 | −0.214 | Nd | −0.383 | 0.243 |
| FeO | −0.017 | −0.505 | Se | −0.561 | 0.155 | Sm | 0.328 | −0.616 |
| MnO | −0.365 | −0.226 | Br | 0.579 | 0.022 | Eu | 0.461 | −0.315 |
| MgO | 0.040 | −0.006 | Rb | 0.059 | −0.391 | Gd | 0.074 | −0.474 |
| CaO | −0.275 | 0.551 | Sr | −0.448 | 0.881 | Tb | 0.31 | −0.479 |
| $Na_2O$ | 0.924 | 0.048 | Zr | 0.207 | −0.040 | Yb | 0.087 | −0.140 |
| $K_2O$ | 0.427 | −0.437 | Mo | 0.334 | −0.021 | Hf | 0.375 | −0.364 |
| Cl | 0.999 | 0.036 | Sn | 0.464 | 0.304 | Ta | 0.207 | −0.419 |
| Sc | 0.110 | −0.426 | Sb | 0.833 | 0.180 | W | 0.369 | 0.140 |
| V | 0.019 | −0.070 | I | 0.247 | 0.427 | Th | 0.046 | −0.282 |
| Cr | −0.321 | 0.186 | Cs | 0.075 | −0.563 | U | −0.27 | 0.511 |
| Co | 0.171 | −0.222 | Ba | 0.345 | 0.356 | | | |

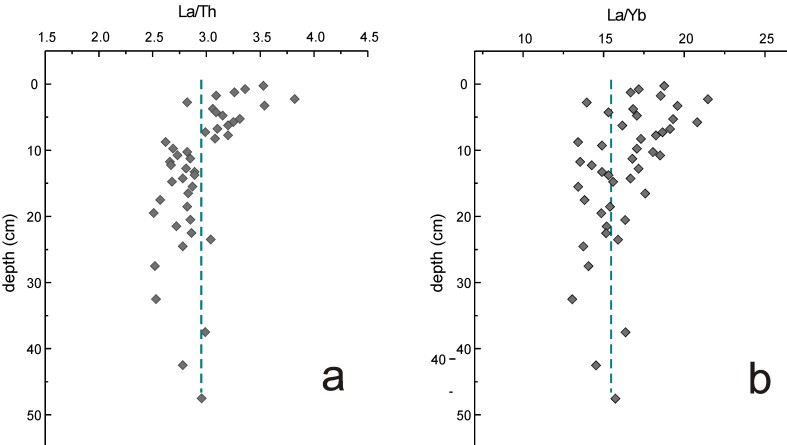

**Figure A3.** The vertical profiles of the La/Th (**a**) and LaYb (**b**) ratios: The vertical green dotted line corresponds to 2.95 (**a**) and 15 (**b**) UCC [30] values of the La/Th and La/Y ratios.

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
