# Peer review of "The Geochemistry of 1 ky Old Euxinic Sediments of the Western Black Sea"

_geosciences, doi:10.3390/geosciences9110455_

Round 1

Reviewer 1 Report

Overall, this could be an important contribution of our understanding of the geochemistry of euxinic water masses and their deposits. What this paper lacks is not a unique dataset, but an in depth discussion of the geochemistry of the sediments. Most of the paper is spend summarizing and presenting abundance data, but little effort is done to interpret the observed trends. I would encourage the authors to do a more analysis driven paper. 

line 1 data

line 1-4 break up long run on sentence. 

line 8 , except for Cl and CaO, the elemental abundances

line 10 except for instead of excepting

line 18 maximum depth or average depth? be specific

line 18-19  rephrease: "2,212m, is also the largest .."

line 20: seawater instead of salted water

line 23 dissolved oxygen

figure 1: the text on the figure itself is redundant and needs to be moved into figure caption the core picture needs to be moved to the left and the map insert needs to be increased in size and moved to the right. The photograph of the core and the map insert needs to be enhanced. The map insert shall not overlap with the core. It is critical that the core is visually represented in a way that readers can assess the core.

l 26ff: preplace with: The Black Sea catchment basin extends over Europe and the Anatolian Peninsula and covers an area of 1,874,904 km2. The Western and North-Western basins represent 82% of its entire basin with a total area of 1,520,000 km2.

l27 replace could with can

l 40 delete The in front of Instrumental "Instrumental Neutron Activation Analysis (INAA), in

l 48 for people not versified in these tecnhiques, this needs to be combined with the paragraph on line 40 in order to give the reader an idea what ENAA is and how it compares the INAA

l 55 was core sealed? what temperature?

l59 later being what? days, months? years?

l62 what kind of mineralogical investigations? be specific.

line 71 is an incomplete sentence where word fragments are floating around. Needs to be fixed. " final experimental a regarding" and "together h corresponding"

l76 in methods it needs to be explained how mineralogy was determined

l77 calls coccolithospheres minor then l80 they become abundant- needs to be quantified. whats the percentage?

l85. there is absolutely no presentation of the results of the elemental data. If it is the main goal of this contribution to present new data on the geochemistry of these deposits, then there must be a more lengthy presentation of the results of the analyses. Right now the bulk of the results is taken up by mineralogy and coccolithosphere tests. The authors need to lead the reader through the results before heading to the discussion. I expect the authors to beef up the results part. 

l98 provenience we normally in sedimentary geology use the terms provenance

l102 sentence fragment. fix

l110 almost not almos

l111 add the `"the sediments'

l113 sediments appear, not appears

l122 second not secon 

l129 the present case 

l130 if you explained it, that means the past, did you publish this? if so you need to add a reference. if not, and this is your present explanation for this paper you have to add the reason why and change it to explain

l142ff this is a description of the resutls, there is no discussion of what it means. needs to be evaluated not just mentioned that there is a peculiariety 

l147 ff same again. you describe the results (so wrong section of paper) and then you conclude that "This peculiarity points towards a slow and continuous change of the sediment geochemistry in the last 400 years". Yes, clearly it does. But why? Here in the discussion part one needs to discuss the data and make an interpretation. What is for this ratio and the one in the previous paragraph?

l151: yes, but what drives this tendency. a short discussion on the geochemistry of Se would help to make a case for what the overall reason is for the observed geochemistry. 

l258 citation 23 Rudnick misspelled 

Author Response

First reviewer

Overall, this could be an important contribution of our understanding of the geochemistry of euxinic water masses and their deposits. What this paper lacks is not a unique dataset, but an in depth discussion of the geochemistry of the sediments. Most of the paper is spend summarizing and presenting abundance data, but little effort is done to interpret the observed trends. I would encourage the authors to do a more analysis driven paper.

Remark

line 1 data

Answer

Corrected

Remark

line 1-4 break up long run on sentence.

Answer

Done

Remark

line 8 , except for Cl and CaO, the elemental abundances

Answer

Corrected

Remark

line 10 except for instead of excepting

Answer

Corrected

Remark

line 18 maximum depth or average depth? be specific

Answer

Corrected: maximum depth

Remark

line 18-19 rephrase: "2,212m, is also the largest .."

Answer

Rephrased

Remark

line 20: seawater instead of salted water

Answer

Corrected

Remark

line 23 dissolved oxygen

Answer

We have kept devoid of oxygen (lacking oxygen was less expressive)

Remark

Figure 1: the text on the figure itself is redundant and needs to be moved into figure caption the core picture needs to be moved to the left and the map insert needs to be increased in size and moved to the right. The photograph of the core and the map insert needs to be enhanced. The map insert shall not overlap with the core. It is critical that the core is visually represented in a way that readers can assess the core.

Answer

We have completely redrawn the Figure 1 accordingly. We have also included the vertical profile of main mineral fraction

Remark

line 26: replace with: The Black Sea catchment basin extends over Europe and the Anatolian Peninsula and covers an area of 1,874,904 km2. The Western and North-Western basins represent 82% of its entire basin with a total area of 1,520,000 km2.

Answer

We have replaced. Thank you!

Remark

line 27: replace could with can

Answer

Replaced

Remark

line 40: delete The in front of Instrumental "Instrumental Neutron Activation ….

Answer

Deleted

Remark

line 48: for people not versified in these tecnhiques, this needs to be combined with the paragraph on line 40 in order to give the reader an idea what ENAA is and how it compares the INAA

Answer

We have included: Rows 69-78

In the case of ENAA, the neutron energies varied between thermal 0.025 eV and resonant 500 eV

which significantly increases both sensitivity and accuracy of measurement [22]. At its turn, the PGAA, which is based on the analysis of the gamma rays emitted during neuron capture by nuclei, allows

determining with high accuracy the content of some elements such as Gd, for which classic INAA

gives great errors. At the same times there are more elements such as Na, Al, Cl, K, Ca, Ti, Mn, Fe or

Sm of which content can be determined with comparable accuracy by both methods which allows to check their analytical performances. More details concerning ENAA and PGAA determinations can be

found in [23–25].

Remark

line 55 was core sealed? what temperature?

Answer

We have add: Rows 82-83:

the 12-cm-diameter BS 600 core was, sealed and stored in vertical position in a refrigerator for about a week. Then, after the end of cruise, the core was examined

Remark

line 59 later being what? days, months? Years?

Answer

Row 57: few weeks later

Remark

line 71 is an incomplete sentence where word fragments are floating around. Needs to be fixed. " final experimental a regarding" and "together h corresponding"

Answer

We have replaced by: Rows: 93-97

The final experimental data regarding the content of the eight major oxides (in wt %) and

31 trace elements (in mg/kg), as determined by ENAA and PGAA can be freely accessed at

http://dx.doi.org/10.17632/d5f8c9gtv7.2 [22], while the main statistical descriptors inclusive the

experimental uncertainty defined as the combination of statistical and certified material uncertainties

are provided in Table A2.

Remark

Line 85. there is absolutely no presentation of the results of the elemental data. If it is the main goal of this contribution to present new data on the geochemistry of these deposits, then there must be a more lengthy presentation of the results of the analyses.

Answer

As mentioned before (Rows 93-97)

.The final experimental data regarding the content of the eight major oxides (in wt %) and

31 trace elements (in mg/kg), as determined by ENAA and PGAA can be freely accessed at

http://dx.doi.org/10.17632/d5f8c9gtv7.2 [22], while the main statistical descriptors inclusive the

experimental uncertainty defined as the combination of statistical and certified material uncertainties

are provided in Table A2.

Moreover, we have included in the Appendix Table A1 and A2 which reproduce the mineralogical composition of each segment of core (Tabla A1) as well as average values, corresponding standard deviations, the total experimental uncertainty of the content of all investigated elements resulting by combining the statistic error in determining the intensity of characteristic gamma spectrum line with the corresponding certified value of the reference material (A2).

Remark

line 98 provenience we normally in sedimentary geology use the terms provenance

Answer

We have changed

Remark

line 102 fragment fix it

Answer

We have changed accordingly: Rows 109 -127

The presence of CaCO3 rich coccolithic debris influenced the distribution of major elements by

increasing the relative content of CaO with respect to UCC as the spider diagram reproduced in Fig. 2a

illustrates. For this reason, the relative content of SiO2 and Al2 O3 decreased accordingly which shifted,

on the ternary diagram SiO2 -Al2O3 -Na2O+K2O+CaO (Fig. 2b), the BS600 points towards smaller SiO2 values with respect to corresponding UCC and NASC points. But excepting this influence, the content of all other major elements remained relatively closer to the UCC as the spidergram reproduced in Fig. 2a shows. These finding suggest that, from the point of view of major elements, the sedimentary material of the BS600 core can be regarded as closer to the UCC with an increased fraction of calcium due to the presence of Emiliania huxleyi coccoliths.

In this regard it should be noted that the iron content, as FeO, if normalized to the sum of all other

oxides excepting CaO, gives a value of 0.067 ± 0.007, higher than the corresponding UCC value of

0.052. This suggests a moderate enriched content of iron in BS600 sediments, in good agreement with

[10] and [31].

Remark

line 110 almost not almos

Answer

Corrected

Remark

line 111 add the `"the sediments'

Answer

Corrected

Remark

line 113 sediments appear, not appears

Answer

Corrected

Remark

line 122 second not secon

Answer

Corrected

Remark

line 129 the present case

Answer

Corrected

Remark

line 130 if you explained it, that means the past, did you publish this? if so you need to add a reference. if not, and this is your present explanation for this paper you have to add the reason why and change it to explain

line 142 if this is a description of the results, there is no discussion of what it means. needs to be evaluated not just mentioned that there is a peculiarity La/Th discussion

Remark

line147 if same again. you describe the results (so wrong section of paper) and then you conclude that "This peculiarity points towards a slow and continuous change of the sediment geochemistry in the last 400 years". Yes, clearly it does. But why? Here in the discussion part one needs to discuss the data and make an interpretation. What is for this ratio and the one in the previous paragraph? La/Yb discussion

Answer to both remarks

We have changed: Rows 179-194

A similar peculiarity was evidenced by investigating the vertical distribution of La/Th ratio.

Usually, its value varies between 2.85 for loess [39] and 2.97 for UCC [29]. In our case we have noticed

an average value of 2.95 ± 0.29, consistent with UCC. After a careful examination of its vertical

distribution along the sediment column, we noticed that this ratio is almost constant between 20 and

50 cm and equal 2.8 ± 0.14, but increases to the surface of sediments to almost 3.5 (Fig. A3a). We have

noticed an analogous behaviour also for the La/Yb ratio. Although the average value of La/Yb ratio

was of 13.4 ± 2.1 very close to the 13.6, which is characteristic for the UCC [39], the La/Yb vertical

profile showed increasing values towards the sediment surface (Fig. A3b), mainly due to a sizeable enlargement of the La content while Yb and Th contain remained almost constant along entire core

[22].

Both cases showed a remarkable resemblance to the vertical profiles of anthropogenic Zn,

As, Br, Sn and Sb reported earlier for the same sediments [36] and attributed to the 18th to 20th

century industrial development of the Western European countries where accompanied by massive

deforestation. At the same time, factories and power plants which are frequently located near water

bodies disposed large amounts of industrial waste which transported by tributary rivers, and especially

by Danube have contaminated the superficial layers of Black Sea sediments.

Remark

line 151: yes, but what drives this tendency. a short discussion on the geochemistry of Se would help to make a case for what the overall reason is for the observed geochemistry.

Answer

We have included:Rows 197-203:

Selenium belongs in Group 16 of the periodic table. In see water, and under oxic conditions, Se is

soluble as selenate SeO42- or selenite SeO32-- ions [35]. In a more reducing environment such as euxinic

one, Se occur as insoluble selenide Se2− ion or even as metallic Se which are further incorporated into

sediments [36]. Moreover, the presence of insoluble Se is related to the activity of both aerobic and

anaerobic organism. Therefore, the presence of the euxinic environment together with the activity of

anaerobic bacteria could the increased content of Se in BS600 sediments as well as its characteristic

vertical profile illustrated in Fig. 3b.

Remark

line l62 what kind of mineralogical investigations? be specific.

Remark

line 176 in methods it needs to be explained how mineralogy was determined

Answer to both remarks

We have included: Rows 77-79.

X-ray diffraction performed by a Cu anode DRON-3 diffractometer provided with a graphite monochromator and a NaI(Tl) detector as well as binocular microscopy was used to determine the content of mineral component.

Remark

line l77 calls coccolithospheres minor then they become abundant- needs to be quantified. whats the percentage

Answer

We have included: Rows 116-122.

Excepting this influence, the content of all other major elements remained relatively closer to the UCC as the spidergram reproduced in Fig. 2a shows. These finding suggest that, from the point of view of major elements, the sedimentary material of the BS600 core can be regarded as closer to the UCC excepting the CaO of which content, according to data reproduced in Table A1, appears 7.9 ± 1.2 times higher then the UCC one. The increased content of CaO reflects in fact the carbonates accumulation in sediments as Fig. 1b and XRD data reproduced in Table A2, most probable due to the presence of Emiliania huxleyi coccoliths [31,34].

Remark

Right now the bulk of the results is taken up by mineralogy and coccolithosphere tests. The authors need to lead the reader through the results before heading to the discussion. I expect the authors to beef up the results part.

Answer

We have introduced more experimental data mainly concerning the vertical distribution of the main mineral components of sedimentary material (Tabla A1-appendix A), major and trace element main statistic descriptors (Table A2-Appendix A) as well as the composition of the Principal Components PC1 and PC2 (Table A3-Appendix).

Remark

Line 258 citation 23 Rudnick misspelled

Answer

Corrected

NB Due to some peculiarities of the LATEX template, the firs two table reproduced in Appendix A were marked E1 and E2 instead of A1 and A2.

Reviewer 2 Report

Thank you for the opportunity to review this manuscript. I have given it a thorough technical review. I am submitting a scan of the document with my detailed comments and summarize my major technical comments here. There are many points that need to be addressed before this manuscript is ready for publication, and I do not recommend it for publication in its current form.

The copy of the manuscript as it was presented to me was a single-spaced pdf with many editorial errors. For future submissions, I recommend that the authors use double-spaced format for easier editing, and that they have someone give their document a thorough editorial review. In some cases, there were whole passages of text obviously missing from cutting and pasting, making it difficult to follow the flow of the study. The study uses ENAA and PGAA to measure a large suite of elements and state in the abstract that determinations had an accuracy of 7%. There is no other indication of analytical accuracy in the text and perhaps they meant a precision of 7%, not accuracy? These methods are highly variable for different elements and more detailed information on the accuracy and precision of the methods for each analyte would be required to make any significant conclusions. Lines 71-74 in the Analytical Techniques section seem to be missing whole lines of text, so perhaps some of this was missed here? The study presents results from a single core in the Black Sea. Inferences made on provenance of Black Sea sediments are weak based on only one location. The main conclusion of the study is that the core sediments are similar to UCC and NASC, but the grouping of the sample results are all points from the single core. Furthermore, the UCC and NASC don’t plot with the core results in the ternary plot in figure 2. I indicated in a number of places on the text, where broad conclusions were made on limited evidence. Results of principal components analysis are presented without supporting statistics, making it difficult to determine the validity of the results. Interpretation of the components is not possible without reviewing the loadings on the variables and the significance of the results. This study is not unique. The Black Sea contains a large euxinic environment and sediment geochemistry has been studied extensively in this basin.

Author Response

The second reviewer

Thank you for the opportunity to review this manuscript. I have given it a thorough technical review. I am submitting a scan of the document with my detailed comments and summarize my major technical comments here. There are many points that need to be addressed before this manuscript is ready for publication, and I do not recommend it for publication in its current form.

Remark

The copy of the manuscript as it was presented to me was a single-spaced pdf with many editorial errors.

For future submissions, I recommend that the authors use double-spaced format for easier editing, and that they have someone give their document a thorough editorial review.

Answer

The present revised version is double-spaced format and checked for language

Remark

In some cases, there were whole passages of text obviously missing from cutting and pasting, making it difficult to follow the flow of the study.

Answer

We have tried to correct all of them

Remark

The study uses ENAA and PGAA to measure a large suite of elements and state in the abstract that determinations had an accuracy of 7%. There is no other indication of analytical accuracy in the text and perhaps they meant a precision of 7%, not accuracy?

Answer

We have provided in Table A2 (Appendix a) all statistic data concerning the main statistical descriptors inclusive the experimental uncertainty defined as the combination of statistical and certified material uncertainties

Remark

These methods are highly variable for different elements and more detailed information on the accuracy and precision of the methods for each analyte would be required to make any significant conclusions.

Answer

We have included in the Table A1 the most relevant statistic data concerning each element, i.e. main (average) value, and we have recalculated the standard deviation by taking into account for each sample the contribution of statistic fluctuation of the corresponding lines of the gamma-ray spectrum as well as certified reference sample errors to the total experimental uncertainty.

Remark

Lines 71-74 in the Analytical Techniques section seem to be missing whole lines of text, so perhaps some of this was missed here?

Answer

We have replaced by: Rows: 93-97

The final experimental data regarding the content of the eight major oxides (in wt %) and 31 trace elements (in mg/kg), as determined by ENAA and PGAA can be freely accessed at http://dx.doi.org/10.17632/d5f8c9gtv7.2 [22], while the main statistical descriptors inclusive the experimental uncertainty defined as the combination of statistical and certified material uncertainties are provided in Table A2.

Remark

The study presents results from a single core in the Black Sea. Inferences made on provenance of Black Sea sediments are weak based on only one location. The main conclusion of the study is that the core sediments are similar to UCC and NASC, but the grouping of the sample results are all points from the single core.

Furthermore, the UCC and NASC don’t plot with the core results in the ternary plot in figure 2. I indicated in a number of places on the text, where broad conclusions were made on limited evidence.

Answer

We have changed accordingly: Rows 109 -127

The presence of CaCO3 rich coccolithic debris influenced the distribution of major elements by increasing the relative content of CaO with respect to UCC as the spider diagram reproduced in Fig.2a illustrates. For this reason, the relative content of SiO2 and Al2 O3 decreased accordingly which shifted, on the ternary diagram SiO2 -Al2O3-Na2 O+K2O+CaO (Fig. 2b), the BS600 suggests smaller SiO2 values with respect to corresponding UCC [29] and NASC [30] points. For this reason, the BS600 data are confined within an elongated ellipse, so higher the CaO content, more distant with respect to UCC and NASC they appear on diagram.

Excepting this influence, the content of all other major elements remained relatively closer to the UCC as the spidergram reproduced in Fig. 2a shows. These finding suggest that, from the point of view of major elements, the sedimentary material of the BS600 core can be regarded as closer to the UCC excepting the CaO of which content, according to data reproduced in Table A1, appears 7.9 ± 1.2 times higher then the UCC one. The increased content of CaO reflects in fact the carbonates accumulation in sediments as Fig. 1b and XRD data reproduced in Table A2, most probable due to the presence of Emiliania huxleyi coccoliths [28,31].

In this regard, it should be noted that, the iron content, as FeO, if normalized to the sum of all other oxides excepting CaO, gives a value of 0.067 ± 0.007, higher than the corresponding UCC value of 0.052. This suggests a moderate enriched content of iron in BS600 sediments, in good agreement with literature data for euxinic basin margin [10] but smaller than those recorded for deep euxinic basin [33] (Table 1).

Remark

Results of principal components analysis are presented without supporting statistics, making it difficult to determine the validity of the results.

Interpretation of the components is not possible without reviewing the loadings on the variables and the significance of the results.

Answer

The loading factors for the Principal Component 1 and 2 are reproduced in Table A3.

In this regard, we have introduced the following comments Rows 208-218:

(Appendix A). In the case of PC1 which assures 38.5 % of total variability, the main contributors are Na2 O, Cl, and K2O, i.e. elements associated with the marine environment and which, as mentioned before form a cluster (Fig. 1a) and possible anthropogenic contaminants Zn, As, Br, Sn, Sb as well La, i.e. those element of which content significantly increased toward sediment surface [34] and Fig. A3. It should be remarked the reduced contribution to PC1 of the SiO 2, TiO−2, Al−2O−3, FeO, MgO, i.e. the major of components of sedimentary material of which presence could be attributed to argillaceous fraction, suggesting the relative uniformity of the mineralogical composition of BS600 sedimentary material.

In the case of PC2, the main contribution comes from CaO and Sr, two components mainly associated with the carbonate fraction of sediments.

Remark

This study is not unique. The Black Sea contains a large euxinic environment and sediment geochemistry has been studied extensively in this basin.

Answer

We totally agree. Our study concerns relatively young the euxinic sediments collected at a depth 600 m, on the slope of continental platform in the vicinity of the Danube Delta where we expect a certain influence of the fresh sedimentary material transported by the Danube river.

We have included Rows 161-172

We explained this finding by the position of BS600 collecting point, on the slope of Continental Platform, while the literature data [10,12,31] refer to the Black Sea abyssal plain where the euxinic water column is significantly thicker.

This assumption is sustained in the case of Fe of which EF is comparable with similar values reported in [10] for euxinic margin, but smaller than those reported for euxinic basin [31]. for two deep stations located in the central part of the Black Sea where the euxinic water column reaches maximum values.. A similar situation we have found in the case of Mo, of which EF, in the case of BS600, reached a maximum value of 60, smaller then the minimum value of 80 reported in [31] too.

To this situation could also contribute the fine suspended sedimentary material transported by the Danube river that could reach the BS600 sampling point situated on the slope of the Continental Platform at a distance smaller than 200 km from the Daube Delta (Fig. 1 - inset).

NB Due to some peculiarities of the LATEX template, the firs two table reproduced in Appendix A were marked E1 and E2 instead of A1 and A2.

Reviewer 3 Report

Summers are pretty busy and I’ve only had bits and pieces of time to look at the manuscript. In any case, it wasn’t going to be a good review and not just because of the problems with the writing / organization.

They require more information about the methods, in particular they need to state detection limits and/or errors.

They also need to consider that the chance in their sampling resolution at ~ 20 cm might be influencing what is observed.

The discussion about Na and Cl can be removed (…these are marine sediments so it is no surprise that these elements are high).

The distribution of redox sensitive trace metals is interesting, particularly Se, but they barely talk about it and provide next to no background information. There is also no discussion of the lack of a Ce anomaly, something I was expecting and while they note the presence of a negative EU anomaly they don’t explain why it is there.

The paper needs to be much more rigorous in its discussion of the data and they need to focus more on the redox sensitive trace metals and less on the the detrital mineralogy.

Author Response

The third reviewer

Summers are pretty busy and I’ve only had bits and pieces of time to look at the manuscript. In any case, it wasn’t going to be a good review and not just because of the problems with the writing / organization.

Remark

They require more information about the methods, in particular they need to state detection limits and/or errors.

Answer

We have included more details explaining the analytical methods we have used: Rows 69-79

In the case of ENAA, the neutron energies varied between thermal 0.025 eV and resonant 500 eV which significantly increases both sensitivity and accuracy of measurement [23]. At its turn, the PGAA, which is based on the analysis of the gamma rays emitted during neuron capture by nuclei, allows determining with high accuracy the content of some elements such as Gd, for which classic INAA gives great errors. At the same times there are more elements such as Na, Al, Cl, K, Ca, Ti, Mn, Fe or Sm of which content can be determined with comparable accuracy by both methods which allows to check their analytical performances. More details concerning ENAA and PGAA determinations can be found in [24–26].

The final experimental data regarding the content of the nine major oxides (in wt %) and 31 trace elements (in mg/kg), as determined by ENAA and PGAA can be freely accessed at http://dx.doi.org/10.17632/d5f8c9gtv7.2 [22].

X-ray diffraction performed by a Cu anode DRON-3 diffractometer provided with a graphite monochromator and a NaI(Tl) detector as well as binocular microscopy were used to determine the content of mineral component.

Remark

They also need to consider that the chance in their sampling resolution at ~ 20 cm might be influencing what is observed.

Answer

In the Table A2 (E2 on manuscript) we have provided the complete vertical profile of main mineral components of the core. Here, the sampling rate was of 5 mm for the first 150 mm, 10 mm from the next 100 mm and 50 mm for the rest 250 mm of the core. According to literature data, we have used one of the highest sampling rate reported until present.

This allowed us to determine the 201Pb vertical profile do determine with high accuracy the sedimentation rate, and consequently to estimate de age of entire core sediments.

Remark

The discussion about Na and Cl can be removed (…these are marine sediments so it is no surprise that these elements are high).

Answer

We have restrained the discussion, but we have not removed it as the Na2O and Cl vertical profiles suggest a partial removal of interstitial water by sediments compaction: Rows 128-137

Concerning the interrelation between the oxides of major components and Cl, the correlation analysis shows the existence of at least three clusters (Fig. 2a, inset). One of them which consists of Na, K and Cl, is most probably due to the presence of interstitial sea water in sediments of which content decreases towards core bottom by sediments compaction, hypothesis sustained by the almost coincident vertical profile of Na2 O and Cl (Fig. A1). The second cluster contains Al, Si, Ti and Fe, which can evidence an association with the clay while calcium oxide, the unique member of the third cluster, negatively correlates with all others (Fig. 2a, inset). As Al, Si, Ti and Fe usually are associated with the argillaceous fraction of sedimentary material, this peculiarity is consistent with the CaCO3 distribution, total complementary to the clay, the other major component of sediments, as mentioned before.

Remark

The distribution of redox sensitive trace metals is interesting, particularly Se, but they barely talk about it and provide next to no background information. There is also no discussion of the lack of a Ce anomaly, something I was expecting and while they note the presence of a negative EU anomaly they don’t explain why it is there.

Answer

We have rewritten this section: Rows 228-238

This finding could be confirmed by the distribution of eight REE as determined by ENAA (La, Ce, Nd, Sm, Eu, Tb and Yb) and PGAA (Gd) and normalized to chondrite [40] (Fig.4c). This digaram evidence no negative Ce anomaly as would expected in the case of euxinic/reducing conditions but a well represented negative Eu anomaly [15] described by an average value of EuN /EuN∗ of 0.6 ± 0.2, close to 0.67 for UCC [29].

The absence of Ce negative anomaly could be related to the relative reduced depth of anoxic/euxinic water column of only 450 m as well as, as mentioned before, to the vicinity of Danube Delta through which the majority of fresh sedimentary material is discharged into Black Sea. In the absence of more literature data on Ce anomaly, further research concerning different region of the Black Sea would be necessary.

While Ce anomaly is determined by the oxic-anoxic condition of depositional environment, Eu negative anomaly is a characteristic for UCC [29] as well as recycled sedimentary rocks [15] such as NASC [30] or PAAS [39]. This fact confirms once more the contribution of terrigenous continental material to BS600 sediments.

Remark

The paper needs to be much more rigorous in its discussion of the data and they need to focus more on the redox sensitive trace metals and less on the the detrital mineralogy.

Answer

We have included more comments concerning the mineralogy, Fe and Se presence as well as the result of Principal Component Analysis together with the explanation of La/Th and La/Yb verical profiles.

NB Due to some peculiarities of the LATEX template, the firs two table reproduced in Appendix A were marked E1 and E2 instead of A1 and A2.

Round 2

Reviewer 2 Report

This version was much improved over the last, but still needs a fair amount of editing. I have attached my edit of the updated version. My main technical comments are that the Spearman's correlation table in Figure 2 is not valid because the samples are from the same core and autocorrelated, and that I doubt that your conclusion that the core is consistent with UCC (lines 179-181) is statistically significant. There is too much overlap of the values.

Author Response

Remark

This version was much improved over the last, but still needs a fair amount of editing. I have attached my edit of the updated version. My main technical comments are that the Spearman's correlation table in Figure 2 is not valid because the samples are from the same core and autocorrelated, and that I doubt that your conclusion that the core is consistent with UCC (lines 179-181) is statistically significant. There is too much overlap of the values.

Answer

We have changed the Fig. 2a by removing the Spearman's correlation table. In our opinion, the correlation table express the existing association between main oxides which in fact reflects the presence of two main mineralogical components: clay and carbonates together with a significant presence of salted marine water.

On the other hand, we totally agree that the correlation table does not prove that the BS600 sediments are consistent with UCC.

For this reason we have removed from the manuscript any statements about the association between correlation table and the UCC, but we kept the table as Table 2 considering that the association between different oxides worth being presented and discussed as an intrinsic characteristic of sediments.

Author Response

Reviewer III

Version 2 of this manuscript is an improvement over version 1 but there are still numerous

edits that need to be made to improve the writing.

Remark

Line 1 – needs a comma after sediments (e.g., sediments, a 50 cm core…)

Answer

Corrected

Remark

Line 3 – Is the “and” needed or should it read “Prompt Gamma, Epithermal Neutron Activation

Analysis”

Answer

Yes, we kept it

Remark

Lines 5 and 10 – If your list is correct then there were 33 (not 32) trace elements analyzed.

Answer

Corrected, Mn appeared twice

Remark

Line 7 – I suggest moving the sentence about 210Pb dating to line 3 (e.g., The core contained

unconsolidated sediment rich in coccolithic ooze and mud that previous 210Pb geochronology

suggest are ~ 1 kyr old.)

Answer

Corrected accordingly: Row 8

Remark

Line 11 – Remove the “of” and “contents” (e.g., “... Fe, Se, Mo, and U, which were

significantly...”)

Answer

Corrected

Remark

Lines and 12 – I have no idea what you mean by “fluctuated in a reduced interval of value” and

what that has to do with the stability of euxinic conditions. Please clarify.

We have removed from text

Remark

Line 18 – Replace “an are” with “a surface area”.

Answer

We have replaced

Remark

Line 22 – Replace “Sea” with “basin”.

Answer

We have replaced

Remark

Line 23 – a comma is needed after the word “sulfide” (e.g., hydrogen sulfide, forms ...).

Answer

We have corrected

Remark

Lines 26 and 27 – Remove the sentence starting with “The Western and North-Western...”. It is

not necessary.

Answer

We have removed

Remark

Lines 27 and 28 – Rewrite as follows “... with a combined inflow of 261 km3/y, representing

76% of the entire tributary discharge into the Black Sea.”

Answer

We have rewritten it - Rows 29-30

Remark

Lines 29 and 30 – Remove “a long section of” from this sentence and replace “on the euxinic

zone” with “within the euxinic zone”.

Answer

We have removed it

Remark

Lines 31 to 33 – This sentence should be moved to the end of line 24 on the previous page.

Answer

We have moved it – Rows 25-27

Remark

Lines 34 to 37 – Move this sentence to the end of Line 46 below. Replace “incompatible and

insoluble elements” with “other elements”.

Answer

We have moved it – Rows 34-37

Remark

Lines 38 to 40 – Move this sentence to the beginning of section 2.2.

We have moved it – Rows 62-64

Remark

Line 46 – Remove “by”

Answer

We have removed it

Remark

Line 52 – Remove “city”

Answer

We have removed it – Row 50

Remark

Line 58 – You need to explain that the high-resolution 5 mm scale sampling occurred in the

upper portion of the core and the low-resolution 5 cm scale sampling occurred in the lower

portion of the core. I am also interested in why you changed the sampling resolution. It makes

interpretation of the data more difficult. For example, is the change you see at about 25 cm

real or simply due to the change in sampling resolution.

Answer

It was a compromise between the highest sampling ratio of 5mm and the maximum number of samples which could be experimentally measured within this project. Excepting XRD, TOC and microscope determination, all other measurements were performed in Hungary and Russian Federation which restrained us to submit maximum 45 samples for INAA.

Remark

Line 60 – What mineralogical analysis? Do you mean XRD analysis and microscopy?

Line 62 – Add in lines 38 to 40 here.

Remark

Line 65 – The bracket after “sediment” is not needed.

Answer

We have repmoved it – Row 68

Remark

Line 70 – Remove “At its turn” from the start of the sentence.

Answer

We have removed

Remark

Line 73 – Remove “At the same time there are more” and start the sentence with “Elements...”.

Also, replace “or” with “and”.

Answer

We have corrected – Rows 73 - 76

Remark

Line 74 – Remove “of which content” and add a comma after the word “method”.

Answer

We have corrected

Remark

Line 79 – Please explain how you used XRD to quantify the amount of mineral components.

Also, add an “s” to “component”.

Answer

We have explained – Rows 77-80

Remark

Line 82 – Remove “by means of a percent stacked area graph”. Also, add a comma before

“while” (e.g., ...illustrated in Fig. 1b, while the corresponding....).

Answer

We have removed it – Row 85

Remark

Line 83 – Remove “At a” from the start of the sentence.

Answer

We have replaced by – Row 85 “A carefully examination of both Fig. 1b and Table A2 showed...”

Remark

Line 84 – Remove the “s” from “carbonates” and the “ing” from “accounting”.

Answer

We have removed it everywhere in the text

Remark

Line 86 – Remove “or” from this sentence and replace it with a comma.

Answer

We have removed it

Remark

Line 88 – You do not need to start a new paragraph here. Also, you should never have

paragraphs that are only one sentence long.

Answer

We have done. Thank you for advise.

Remark

Line 88 to 90 – Rewrite this sentence as follows: “It should also be noted that in the upper

portion of the core carbonate content decreases while quartz was more abundant (Fig. 1b).”

Remark

Line 91 – How was the TOC measured? Please explain this in the methods section.

Answer

We hae explained – Rows - 81-82

Remark

Lines 98 to 101 – This section of the text should be moved to the end of the introduction (i.e.,

after line 46).

Answer

We have moved – Rows - 42-45

Remark

Line 103 – Replace “ data presented by [20]” with “analysis” and add [20] to the end of the

sentence.

Answer

We have changed – Rows – 98 - 99

Remark

Line 105 – You need a space between Fig. and 1, replace “method” with “data” and “to” with

“for” in this sentence.

Answer

We have corrected – Rows - 99-100

Remark

Line 106 – Remove “as”

Answer

We have corrected - Row - 101

Remark

Line 109 – There should be a “-“ between CaCO3 and rich.

Answer

We have corrected – Row – 108

Remark

Line 110 – Remove “as the spider diagram reproduced in Fig. 2a illustrates” and replace it with

“(Fig. 2a)”. Do the same thing for Line 117.

Answer

We have removed – Rows – 109 and 116

Remark

Lines 110 to 115 – I believe all you are trying to say is that due to due to the higher CaO content

BS600 sediments are shifted on the ternary diagram (Fig. 2a). However, this paragraph is

poorly written, with many grammatical errors, and thus confusing. Please clarify.

Answer

We have changed accordingly – Rows 112 - 115

Remark

Lines 116 to 122 – This paragraph is also poorly written and I’m not sure what point you are

trying to make. Please clarify.

Answer

We have changed accordingly – Rows 116 - 120

Remark

Line 126 – Add “the” before “euxinic” in both parts of this sentence and replace “smaller” with

“lower”.

Answer

We have corrected – Rows 119

Remark

Line 129 – There are only two clusters. The third is not a cluster since it is only one component

(i.e., CaO).

Remark

Lines 129 and 130 – Rewrite as follows “One of them, which consists of Na, K, and Cl, is most

probably due to the presence of interstitial seawater. The content decreases towards the base

of the core due to sediment compaction, a hypothesis sustained by the almost ....”

Answer

We have changed and restrained only to two clusters– Rows 121 - 128

Remark

Line 133 – Replace “which can evidence” with “which is probably associated with the presence

of clays.” Start a new sentence when you being discussing CaO and its relationship to CaCO3

and how it dilutes the other components.

Answer

We have, as mentioned before, rewritten the entire phrase – Rows 121 - 128

Remark

Lines 135 to 137 – You already mentioned Al, Si, Ti, and Fe above. Remove this sentence.

Answer

We removed it